# Research on Stability Control Method of Electro-Mechanical Actuator under the Influence of Lateral Force

**Shuai Wu [1], Yong Zhou [1,2,*], Jianxin Zhang [3,4], Shangjun Ma [2,3] and Yunxiao Lian [1]**

[1] School of Aeronautics, Northwestern Polytechnical University, Xi'an 710072, China; shuai981019@163.com (S.W.); yunxiao0324@163.com (Y.L.)
[2] Xi'an Ding Bai Precision Technology Co., Ltd., Xi'an 710000, China; mashangjun@nwpu.edu.cn
[3] School of Mechanical Engineering, Northwestern Polytechnical University, Xi'an 710072, China; xin5137@126.com
[4] Jiangshan Heavy Industries Research Institute Co., Ltd., Xiangyang 441057, China
[*] Correspondence: yongstar@nwpu.edu.cn

**Abstract:** This paper takes a multi-stage Electro-mechanical Actuator (EMA) as the research object, analyzes the lateral force of the multi-stage EMA in the vertical state, and the overall mathematical model of the multi-stage EMA system. Firstly, a permanent magnet synchronous motor module is built in JMAG according to the engineering requirements. Then, the electrical control part and mechanical transmission part of the multi-stage EMA are established in AMESim, and the ideal motor module in AMESim is replaced with the motor model designed by JMAG to construct the overall model of the multi-stage EMA. The dynamic simulation model of lateral force is established in ADAMS to accurately simulate the impact of wind load on EMA in the actual environment, and this model is introduced into AMESim instead of the lead screw and nut module in AMESim. The improved active disturbance rejection control (ADRC) is used to replace the speed loop and positional loop in the traditional three closed-loop control, and the whole system stability servo control of multi-stage EMA is co-simulated. Finally, the experiment of the designed control method is carried out by LabVIEW. The result of the experiment shows that the multi-stage EMA system can effectively suppress the lateral force under the active disturbance rejection control and ensure the stable operation of the multi-stage EMA system. In addition, the system built by the co-simulation method is closer to real working conditions than the traditional mathematical model. The control parameters in the simulation can be effectively transplanted to the actual system with only minor adjustment to meet the engineering requirements.

**Keywords:** multi-stage EMA; active disturbance rejection control; co-simulation



## 1. Introduction

As an irreplaceable weapon in modern war, the reaction speed of missiles will affect the overall situation of modern warfare. The first step of missile launch is to erect the missile from the horizontal to the launch attitude. This process is called the erection process of the missile. The erection time of the missile generally accounts for 20% to 40% of the total launch time. Studying the rapidity of the erection process of the missile is of great significance to shorten the preparation time for missile launch and improve the survivability of the missile weapon system [1–3]. With the progress of science and technology, EMA with the advantages of light weight, high efficiency, high system reliability and easy maintenance have become the development trend of weapon systems in the future [4,5]. EMAs have been widely used in high-performance industrial fields such as aircraft aileron [6], rudders [7], load loading, helicopter rotor adjustment systems [8], missile steering gear [9], spacecraft [10], large precision machine tools [11], and large erect equipment [12,13]. However, many practical factors (such as cogging torque, load torque, friction torque, and load disturbance) and other nonlinear factors in the closed-loop

servo system of EMAs bring difficulties to the high-performance control target. These factors can be regarded as the generalized interference of EMA systems. If they are not compensated and suppressed, it will greatly reduce the accuracy and stability of system control [14]. ADRC technology can observe the total internal and external disturbances of the system in real time through an extended state observer (ESO) and carry out feedforward compensation. It realizes the combination of dynamic response and disturbance rejection ability and has strong robustness and high control accuracy. It is, therefore, very suitable for EMA control based on permanent magnet synchronous motors [15,16]. Reference [17] presents a design of an extended state observer with sliding mode reaching law to estimate the load torque, and the estimated load torque is used as feedforward compensation to improve the anti-interference ability of the speed loop. Reference [18] proposes a load adaptive double loop drive system based on an improved position speed integrated auto-disturbance rejection controller and a parameter fuzzy self-tuning method. The experiments show that the designed drive system has the characteristics of high positioning accuracy, fast positioning response, and a strong adaptability to load changes.

　　Modern high-tech war requires that missile weapon systems have the ability to launch without support in the field. Therefore, the mobile missile will be affected by the wind load in the process of erecting and launching in the field [19]. Wind load is a kind of load with highly nonlinear characteristics, which will cause vibrations in the launcher and potentially cause errors in the normal operation of the instruments and equipment on the missile and the adjustment of the missile control system [20,21]. According to these unique conditions, the stability control method of EMA is studied, and the displacement response of the launcher due to the action of wind load is analyzed. Ensuring that the missile erection equipment is in a normal working state is essential for launch safety. Furthermore, it has important theoretical research significance and practical engineering application value for missile erection control [22].

　　An EMA is mainly composed of a permanent magnet synchronous motor (PMSM), controller, gear reducer, and a planetary roller screw mechanism. At present, single-stage EMAs have been widely used, but because their transmission mechanism is a single-stage roller screw mechanism or ball screw mechanism, the working stroke is limited by the size of the screw. In the application fields of high thrust, long stroke, radar, large equipment erection frame, special vehicle lifting device and so on [23], sufficient displacement cannot be achieved by a single-stage EMA only. Therefore, multi-stage EMAs came into being. It is particularly important to study the driving technology of multi-stage EMAs [24,25], but the control of multi-stage system is more complex than that of single-stage system. While considering the driving control algorithm, we also need to consider the influence of nonlinear factors of a multi-stage roller screw mechanism on the control stability of the whole system. Figure 1 shows the structure of three-stage EMA.

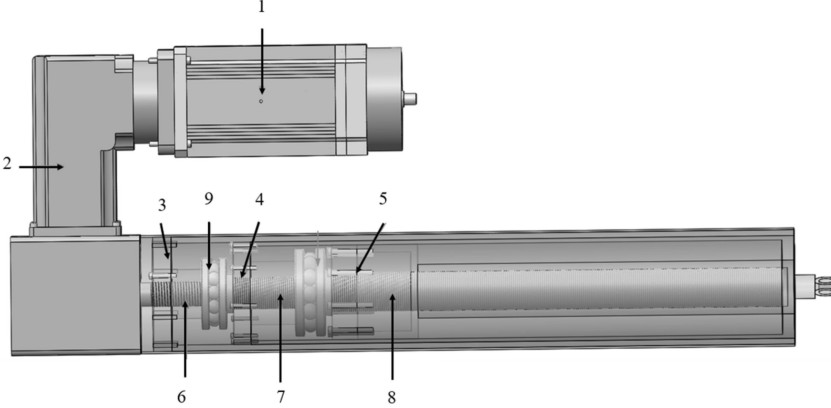

**Figure 1.** Schematic diagram of three-stage EMA. 1-PMSM, 2-Reducer, 3-Primary Nut, 4-Secondary Nut, 5-Tertiary Nut, 6-Primary Main Lead Screw, 7-Secondary Hollow Lead Screw, 8-Tertiary Hollow Lead Screw, 9-Thrust Bearing.

The transmission principle of three-stage EMAs is as follows: Firstly, the rotor of permanent magnet synchronous motor rotates and acts through the reducer to reduce the motor speed and improve the output torque of the motor. Furthermore, the rotation of the primary main screw drives the linear movement of the primary nut. At the same time, the primary nut pushes the secondary planetary roller screw mechanism to move axially, and the primary main screw drives the secondary hollow screw to complete the rotation movement through the hexagonal surface connection at the right end. The secondary hollow screw is driven by the hexagonal surface structure to rotate along with the primary main screw and drives the secondary nut to move in translation. Similarly, the secondary nut drives the overall axial movement of the three-stage planetary roller screw mechanism and drives the tertiary hollow screw to rotate coaxially through the profile structure, so as to finally realize the transmission of speed and force.

Taking a three-stage EMA as the research object, this paper studies the stability control of the three-stage EMA under the influence of lateral force through the improved ADRC technology. The simulation and experiment verify that the ADRC technology designed in this paper has good stability and anti-interference and can meet the operating conditions of the three-stage EMA system under lateral loading. The system has good static and dynamic performance.

## 2. Modeling and Simulation of EMA

In order to simulate the actual working conditions more accurately, different software are used to design the servo drive module, mechanical transmission module, ADRC parts, and lateral force module of the multi-stage EMA erecting device.

### 2.1. Multistage EMA Motor Model Based on JMAG

According to the indicators required by the motor in practical engineering applications, the design motor parameters are shown in Table 1. The built-in rotor is selected, and the pole slot is configured as 8 poles and 54 slots. The motor shape and winding mode are shown in Figure 2.

**Table 1.** Main parameters of motor.

| Parameter | Value |
| --- | --- |
| DC Bus Voltage | 270 V |
| Rated power | 6 kW |
| Rated torque | 19 Nm |
| Peak torque (Low speed) | 50 Nm |
| Rated speed | 3000 rpm |
| Maximum speed | 3500 rpm |
| Rated current (I) | 27 A |

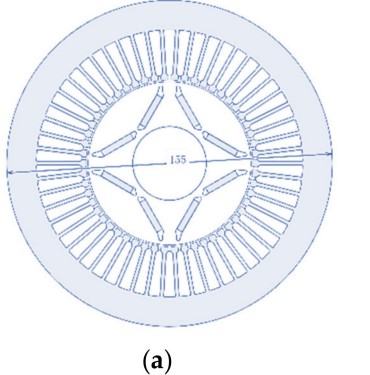　　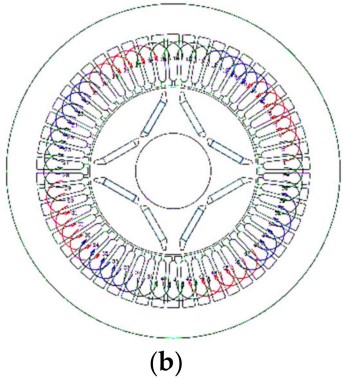

(**a**)　　　　　　　　　　　　　　　　　　　　(**b**)

**Figure 2.** (**a**) Motor shape; (**b**) Winding mode.

The motor operates at the rated speed, and the line voltage waveform is shown in Figure 3a. The sinusoidal degree of the line voltage waveform is relatively good, the effective value of the voltage is 129 V, and the maximum value is 182 V. Fourier analysis of the voltage waveform is shown in Figure 3b. It should be noted that the high-order harmonics are ignored in the Fourier analysis process, which is only used to detect that the peak-to-peak value of the voltage does not exceed the DC bus voltage.

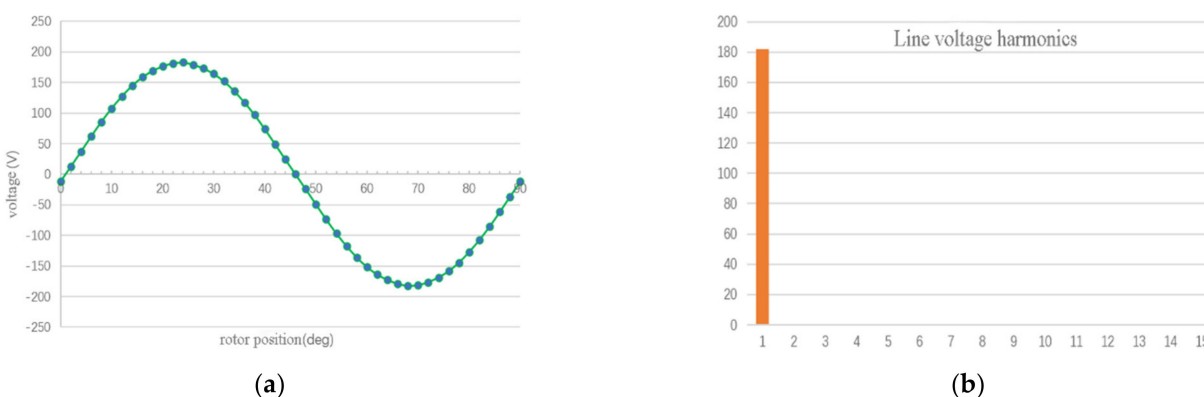

(**a**)         (**b**)

**Figure 3.** (**a**) Line voltage waveform; (**b**) Harmonic component of line voltage.

The motor operates at a rated load, the speed is 3000 rpm, and the effective value of phase current is 27 A. At this time, the magnetic density cloud diagram is shown in Figure 4a. Except for the position of rotor magnetic bridge, the magnetic density and saturation degree of other positions are small. The torque waveform is shown in Figure 4b. When the phase current is 27 A, the average torque value is 19.2 Nm, the torque fluctuation rate is only 1.35%, and the output torque is very stable.

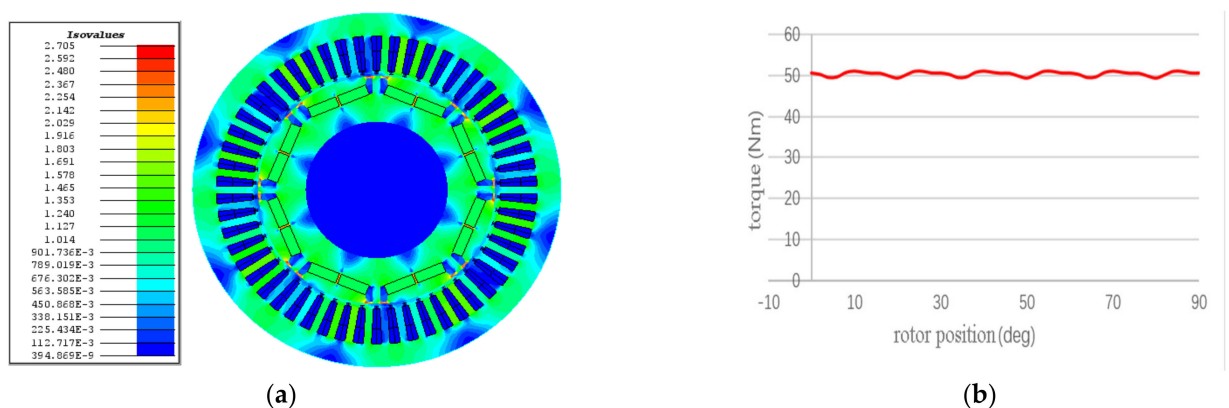

(**a**)         (**b**)

**Figure 4.** (**a**) Load magnetic density cloud; (**b**) Torque waveform.

Under the condition of rated current and rated speed, electromagnetic torque, motor efficiency, power, copper loss, and other indicators are shown in Figure 5.

It can be seen from Figure 5 that the electromagnetic torque of the designed motor under rated current is about 19 Nm and the power is about 6 kW. Meeting the performance indexes in Table 1.

The current phase is divided by 15 degrees from 90 degrees to −90 degrees. The calculated inductance distribution and the relationship between electromagnetic torque and $I_d$, $I_q$.

As can be seen from Figure 6, the inductance and flux linkage of the motor model designed according to the actual parameters in JMAG will change with the current value and phase angle. Therefore, in the subsequent co-simulation, the JMAG motor model is used to replace the ideal motor model with constant inductance and flux parameters in

AMESim that can improve the reliability of the simulation and make the conclusion of more practical value.

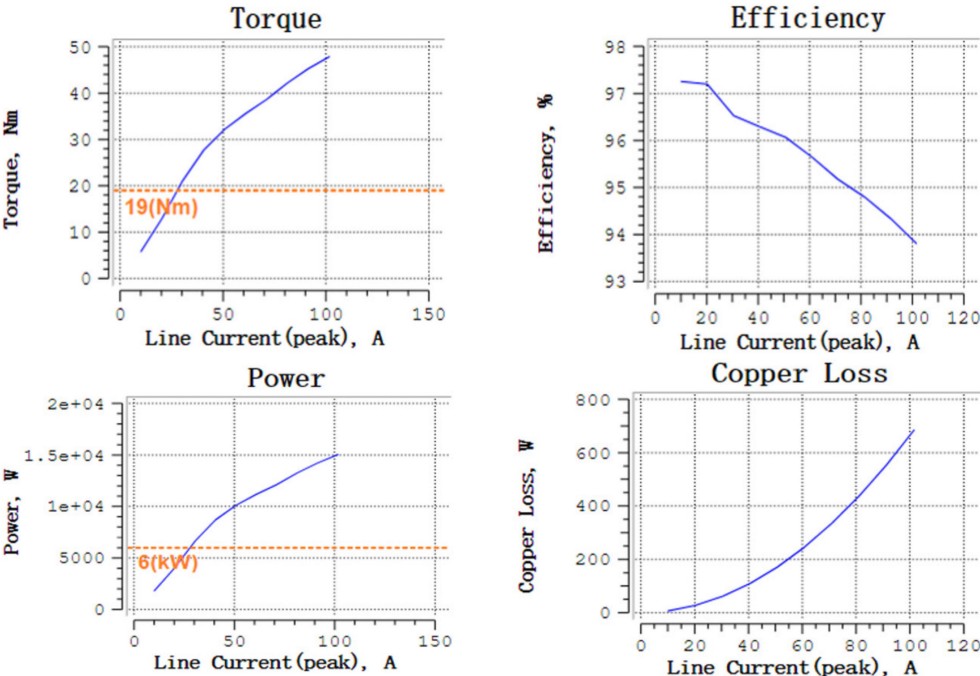

**Figure 5.** Motor performance index.

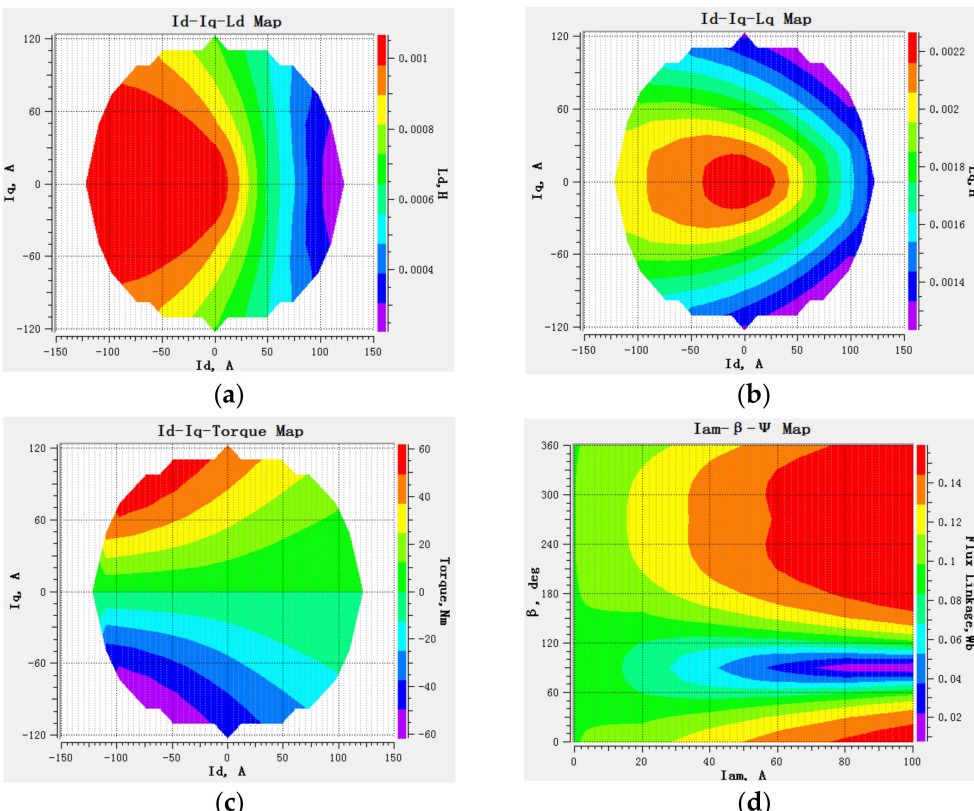

**Figure 6.** Distribution diagram of various parameters of motor model: (**a**) $I_d$ distribution map; (**b**) $I_q$ distribution map; (**c**) Electromagnetic torque distribution map; (**d**) Flux linkage distribution map.

### 2.2. Three-Stage EMA Mechanical Connection Model

According to the structure and principle of the multi-stage EMA mentioned above, the PMSM is connected with the reducer, the reducer is connected with the multi-stage planetary roller screw mechanism, and the multi-stage planetary roller lead screw is connected with the missile body. The overall model of the multi-stage EMA is built in AMESim. Figure 7 shows the schematic diagram of multi-stage EMA model.

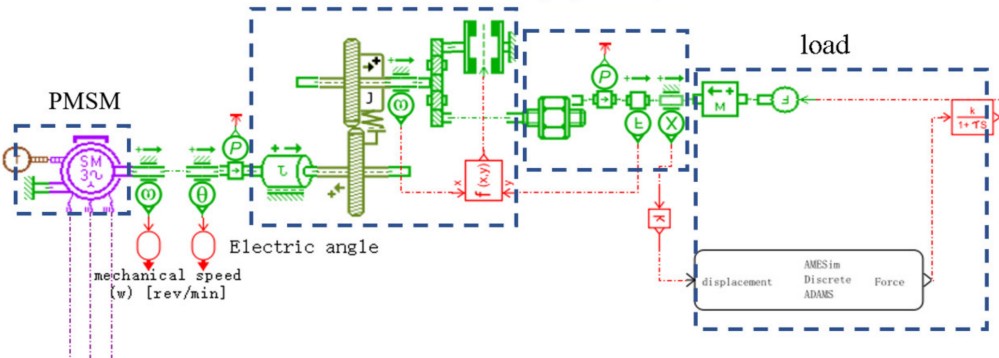

**Figure 7.** Multi-stage EMA model diagram of AMESim.

### 2.3. Model of ADRC

2.3.1. Improved Extended State Observer

Existing first-order system:

$$\dot{y} = f(y,\ w,\ t) + bu \tag{1}$$

In the formula, $y$ is the system state variable, $w$ is the external disturbance, $f(y,\ w,\ t)$ is the total disturbance including external disturbance, and internal disturbance, $b$ is the control gain and $u$ is the control quantity.

Select the state variable $x_1 = y$, $x_2 = f(y,\ w,\ t)$ and write the state equation as follows:

$$\begin{cases} \dot{x}_1 = x_2 + bu \\ \dot{x}_2 = \dot{f} \\ y = x_1 \end{cases} \tag{2}$$

The filter is equivalent to a first-order inertial link, and its expression is:

$$\dot{y}_0 = -ay_0 + ay \tag{3}$$

In the formula, $y_0$ is the output signal of the filter; $a$ is the reciprocal of the time constant of the inertial link.

The output signal of the expansion filter is a new state variable $x_0$, and the state equation of the combined system is:

$$\begin{cases} \dot{x} = Ax + Bu \\ y = Cx \end{cases} \tag{4}$$

Among: $A = \begin{bmatrix} -a & a & 0 \\ 0 & 0 & 1 \\ 0 & 0 & 0 \end{bmatrix}$, $B = \begin{bmatrix} 0 \\ b \\ 0 \end{bmatrix}$, $C = \begin{bmatrix} 1 & 0 & 0 \end{bmatrix}$.

A linear state observer is constructed for this system:

$$
\begin{cases}
\dot{z} = [A - LC]z + [B,\ L]u_c \\
y_c = z
\end{cases}
\tag{5}
$$

$z$ is the observed value of $x$, $L$ is the gain matrix of the observer, $u_c$ is the total input of the observer and $y_c$ is the output of the observer.

Using the pole-placement method, the pole is placed at $-\omega_o$, and the solution can be obtained:

$$
L = \begin{bmatrix} 3\omega_o - a & \dfrac{3\omega_o{}^2}{a} & \dfrac{\omega_o{}^3}{a} \end{bmatrix}^T
\tag{6}
$$

Judge the observability of the system, when $a \neq 0$,

$$
rank \begin{bmatrix} C \\ CA \\ CA^2 \end{bmatrix} = \begin{bmatrix} 1 & 0 & 0 \\ -a & a & 0 \\ a^2 & -a^2 & a \end{bmatrix} \equiv 3
\tag{7}
$$

Therefore, this system is completely observable and can carry out arbitrary pole assignment.

The improved extended state observer retains the advantages of a linear extended state observer, with only one adjustment parameter and simple adjustment. At the same time, the influence of the filter introduction on the system is solved by introducing a new extended state. Time-delay systems are often treated as inertial links, and the improved extended state observer also plays a role in improving the time-delay of the system.

### 2.3.2. Design of ADRC for Multi-Stage EMA

After adopting current closed-loop control, the mathematical model of PMSM is:

$$
\begin{cases}
\dot{\theta}_m = \omega_m \\
\dot{\omega}_m = \dfrac{n_p{}^2 \psi_f}{J} i_q - \dfrac{T_l}{J}
\end{cases}
\tag{8}
$$

$\theta_m$ is the mechanical angle of the motor, $\omega_m$ is the mechanical angular speed, $n_p$ is the number of poles, $\psi_f$ is the permanent magnet flux, $J$ is the moment of inertia, $i_q$ is the q-axis component of the stator current, and $T_l$ is the load torque.

1.　Design of ADRC for speed loop

The system expression of speed loop is:

$$
\dot{\omega}_m = \frac{n_p{}^2 \psi_f}{J} i_q - \frac{T_l}{J}
\tag{9}
$$

Let $x_1 = \omega$, $x_2 = -\dfrac{T_l}{J} = f$, $b = \dfrac{n_p{}^2 \psi_f}{J}$, $u = i_q$, the time constant of inertia link is $a_1$. So, with proportional feedback, the speed loop ADRC is:

$$
\begin{cases}
\dot{z}_0 = -3\omega_o z_0 + a_1 z_1 + (3\omega_o - a_1)y \\
\dot{z}_1 = -\dfrac{3\omega_o{}^2}{a_1} z_0 + z_2 + bu + \dfrac{3\omega_o{}^2}{a_1} y \\
\dot{z}_2 = -\dfrac{\omega_o{}^3}{a_1} z_0 + \dfrac{\omega_o{}^3}{a_1} y \\
u_0 = k_p(v - z_1) \\
u = \dfrac{u_0 - z_2}{b}
\end{cases}
\tag{10}
$$

$z_0, z_1, z_2$ are the observed values output by the observer, $k_p$ is the proportional coefficient, and $v$ is the target value of speed.

2.　Design of ADRC for position loop

The system expression of the position ring is:

$$\dot{\theta}_m = \omega_m \tag{11}$$

Let $x_1 = \theta_m$, $x_2$ is an unknown disturbance, $b = 1$, $u = \omega_m$, the time constant of inertia link is $a_2$.

The same principle as the speed loop, the position loop ADRC is:

$$
\begin{cases}
\dot{z}_0 = -3\omega_o z_0 + a_2 z_1 + (3\omega_o - a_2)y \\
\dot{z}_1 = -\frac{3\omega_o^2}{a_2}z_0 + z_2 + bu + \frac{3\omega_o^2}{a_2}y \\
\dot{z}_2 = -\frac{\omega_o^3}{a_2}z_0 + \frac{\omega_o^3}{a_2}y \\
u_0 = k_p(v - z_1) \\
u = \frac{u_0 - z_2}{b}
\end{cases} \tag{12}
$$

In the formula, $v$ is the target value of displacement.

### 2.4. Dynamic Model Based on ADAMS

The large erecting equipment is analyzed as the applied object of multi-stage EMA. Figure 8a shows the erecting process of the erecting equipment and takes this working condition as an example. The three-dimensional model of the missile body and multi-stage EMA is established in ADAMS, and the mass of the missile body is set to 10 t. The ADAMS model of EMA erecting system is shown in Figure 8b.

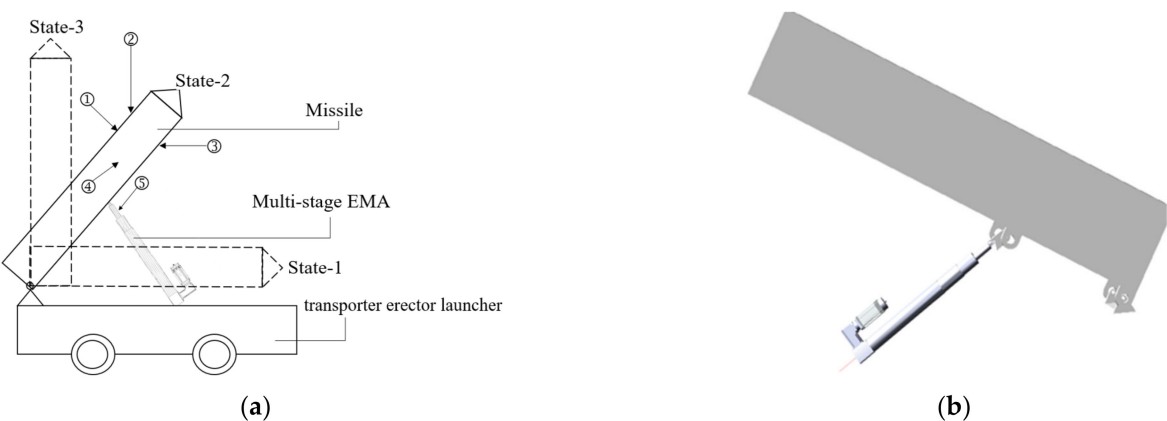

(**a**)  (**b**)

**Figure 8.** (**a**) Schematic diagram of missile erection process; (**b**) ADAMS model of EMA erecting system.

In the Figure 8a, ①, ②, ③ and ④ respectively represent the wind load acting on the missile body, where ① is the force perpendicular to the axial direction of the missile body, which increase the load on the electromechanical actuator. Whereas ②, ③ and ④ are wind loads at different angles to the axial direction of the projectile. Due to the different directions and positions in which they are applied to the missile body, the load force on the electromechanical actuator will increase or decrease. It can increase or decrease the load force on the EMA according to the direction and position acting on the projectile. Lastly, ⑤ is the lateral force acting on the lead screw of EMA, which is at a certain angle with the axial direction of the lead screw. Through decomposition, the oblique lateral force of the EMA can be regarded as two parts: one part is parallel to the extension direction of the EMA, equivalent to the load torque, and the other part is perpendicular to the extension direction, equivalent to the pressure applied on the lead screw. When the lateral force has an uncertain frequency, the component frequency of the different parts cannot be determined.

Figure 9 is a schematic diagram of the wind load acting on the lead screw end of the EMA, in which the component of the lateral force F1 is decomposed in the parallel direction

of the lead screw extension, which is equivalent to the pulling force pulling the lead screw outward, leading to an increase in the speed to erection process. The component of the lateral force F2 decomposed in the extending parallel direction is equivalent to the pressure of pressing the lead screw back, which will reduce the speed to erection process.

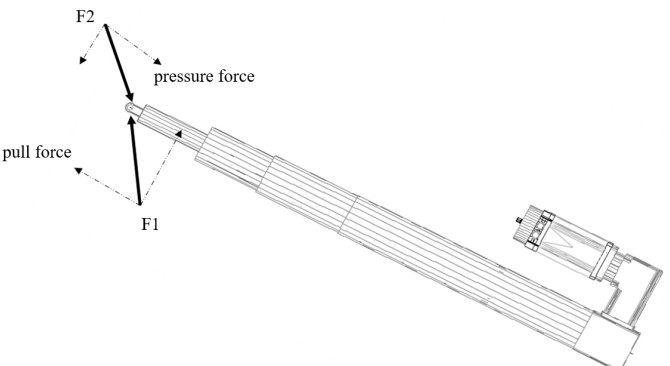

**Figure 9.** Schematic diagram of oblique lateral force decomposition.

## 3. Co-Simulation and Analysis

The dynamic model established in ADAMS to replace the lead screw nut module in the AMESim mechanical library is used and the JMAG motor model is used to replace the ideal motor model in AMESim. Therefore, on the basis of the previous multi-stage EMA overall model, the lead screw nut module is deleted. The speed output from the reducer is transmitted to the speed drive of the primary lead screw of ADAMS though AMESim, and the displacement of the three-stage thrust sleeve measured in ADAMS is taken as the input parameter from ADAMS to AMESim. The displacement value establishes the displacement closed loop in AMESim and calculates the lead screw speed and load force. At the same time, the calculated load force is outputted from AMESim to ADAMS and added to the center of the three-stage thrust sleeve as a point force. The torque calculated from the speed drive of the primary lead screw is transmitted from ADAMS to AMESim to the reducer. The overall parameter transfer relationship is shown in Figure 10.

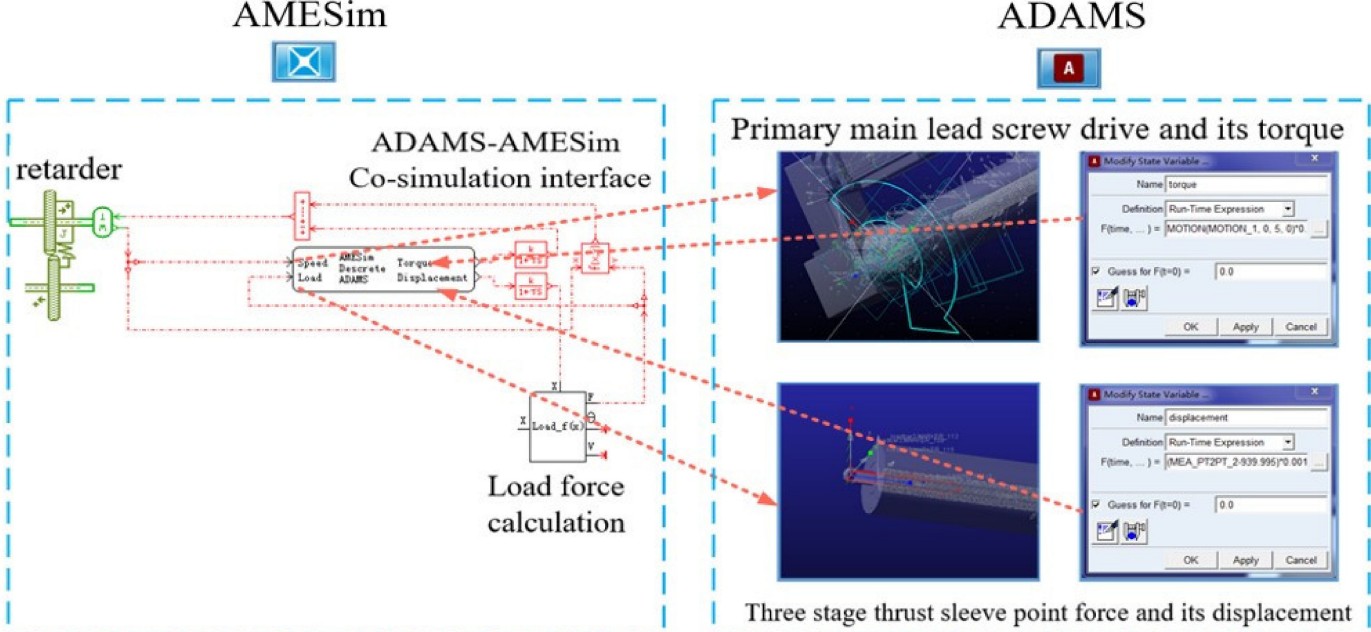

**Figure 10.** Parameter transfer relationship between AMESim and ADAMS.

After determining the parameter transfer relationship, ADAMS is used as the main control and AMESim is used as the attendant control. The specific flow of co-simulation is shown in Figure 11, the overall control block diagram of co-simulation is shown in Figure 12.

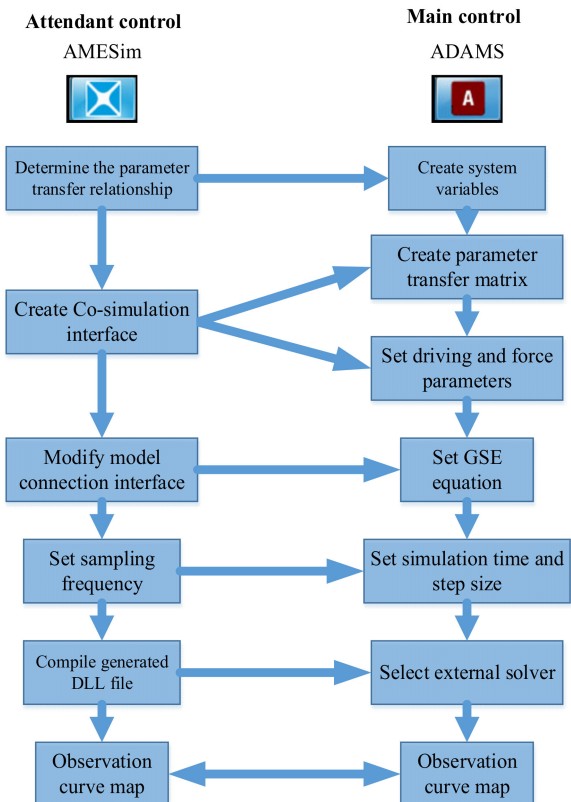

**Figure 11.** Parameter transfer relationship between AMESim and ADAMS (the specific flow of co-simulation).

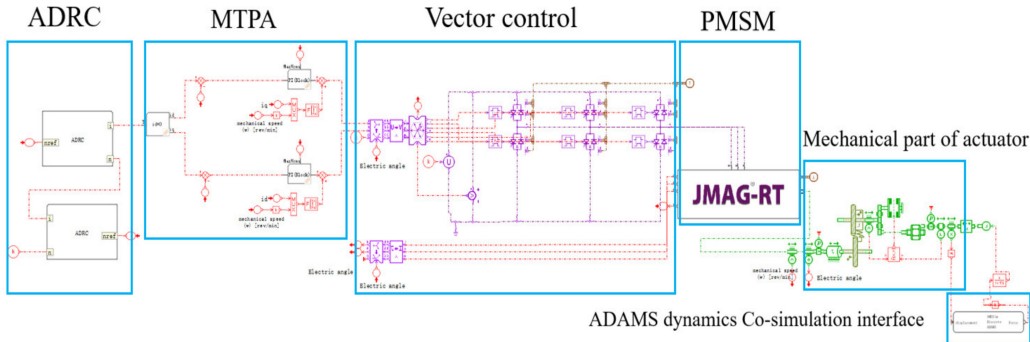

**Figure 12.** Overall control block diagram of Co-simulation.

In AMESim, setting parameters such as DC side voltage $U_{dc} = 270$ V, maximum current $I_{max} = 72$ A, motor pole pair $p = 4$, motor three-phase resistance $R_s = 0.0433$ $\Omega$, and set reducer reduction ratio i $= 24$, sampling time $Ts = 0.0001$ s, moment of inertia $J = 0.00607$ kg.m$^2$.

After setting the simulation parameters, experiments are carried out to verify the effectiveness of the designed algorithm:

### 3.1. Simulation of Stability Control Affected by Impact Lateral Force

In the 20 s of erection, add the lateral force with the size of $1.4 \times 10^3$ N through ADAMS for 10 s, and the angle of the lateral force is 30 ° to the axial direction of the lead screw of the EMA.

The co-simulation results of traditional PID control are shown in Figure 13a,b. The system adjustment time is 25 s, the overshoot is about 5.29%, the motor speed fluctuation amplitude is 1000 rpm, and the displacement is stable at 1.7 m after 25 s. Figure 13c and 13d show the co-simulation results using the ADRC algorithm. It can be seen from Figure 13c that the displacement has no overshoot, and the system reaches stability after 27 s. According to Figure 13d, it can be concluded that the motor speed transitions smoothly without significant fluctuation after sudden lateral force.

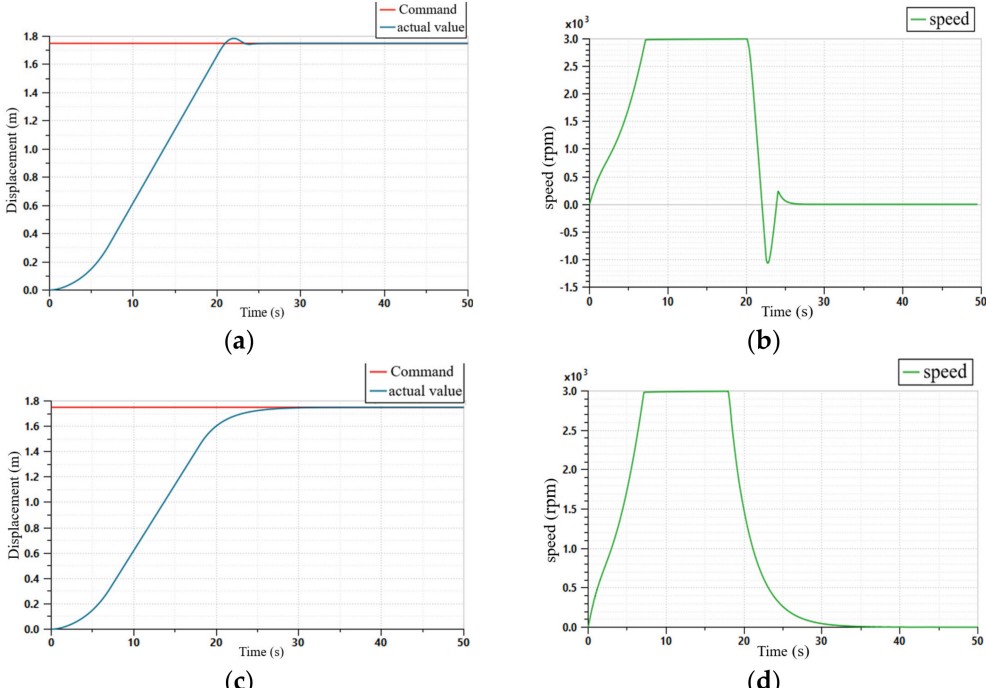

**Figure 13.** Simulation results of the control algorithm under the influence of impact lateral force: (**a**) erecting displacement of PID; (**b**) motor speed of PID; (**c**) erecting displacement of ARDC; (**d**) motor speed of ADRC.

### 3.2. Simulation of Stability Control under the Influence of Periodic Lateral Force

The lateral force frequency is set as $f = 0.25$ Hz, the amplitude is $1 \times 10^3$ N $\sim$ $1.4 \times 10^3$ N, and the direction between the lateral force angle and the axial direction of the lead screw of the EMA is 30°.

Figure 14a is the displacement curve of the EMA under PID control. When the system is affected by periodic lateral force, it reaches a stable state after 45 s, and the displacement amplitude fluctuation after stabilization is about $\pm 7$ cm. Observe the motor speed curve in Figure 14b, after 45 s: the motor speed reaches stability, and the fluctuation amplitude is $\pm 600$ rpm. Figure 14c shows the displacement curve of EMA. The system can reach a stable state after 25 s, and the displacement amplitude fluctuation after stabilization is about $\pm 5$ cm. It can be seen from the speed curve in Figure 14d that the motor speed reaches stability after 25 s, and the fluctuation amplitude is $\pm 1100$ rpm.

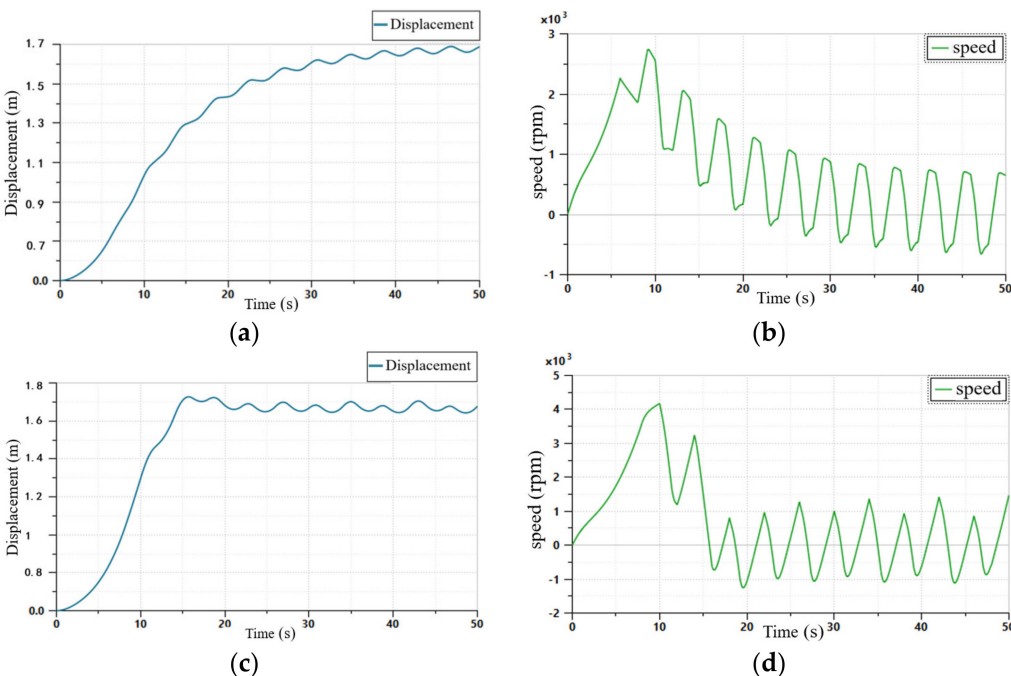

**Figure 14.** Simulation results of the control algorithm under the influence of periodic lateral force: (**a**) erecting displacement of PID; (**b**) motor speed of PID; (**c**) erecting displacement of ARDC; (**d**) motor speed of ADRC.

Through the co-simulation experiment, it can be concluded that the multi-stage EMA erecting system with ADRC algorithm has better anti-interference ability and faster follow-up response ability. Moreover, it can be seen from the erect load force curves in Figure 15a,b that the mass of the missile body is set as 10 t in ADAMS. At the beginning of the erection, the load force on the lead screw end of the multi-stage EMA is $1.1 \times 10^5$ N, which is greater than the gravity of the missile. This is because the lead screw section is affected by the inertia of the missile body. At the moment of the erect stage, the load torque reaches the peak in a very short time, which accurately reflects the impact torque of the load on the EMA at the beginning of the erection. It also embodies the advantages and practicability of co-simulation.

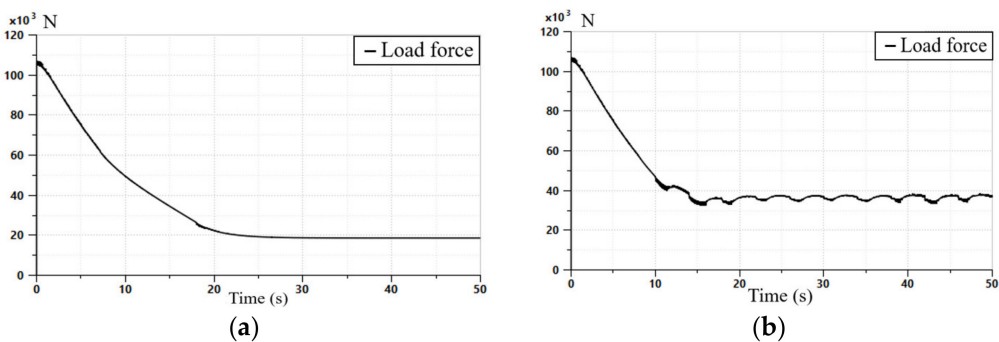

**Figure 15.** (**a**) erect load force curves impact lateral force; (**b**) erect load force curves periodic lateral force.

## 4. Experimental Study on EMA Stability Control under the Influence of Lateral Force

By designing the software and hardware of the multi-stage EMA system experiment platform, the control algorithm is built in LabVIEW, and finally the relevant experiments are carried out.

### 4.1. EMA Servo System and Loading System

The three-stage EMA servo system mainly includes PMSM, drive system and three-stage planetary roller screw mechanism. As shown in the Figure 16.

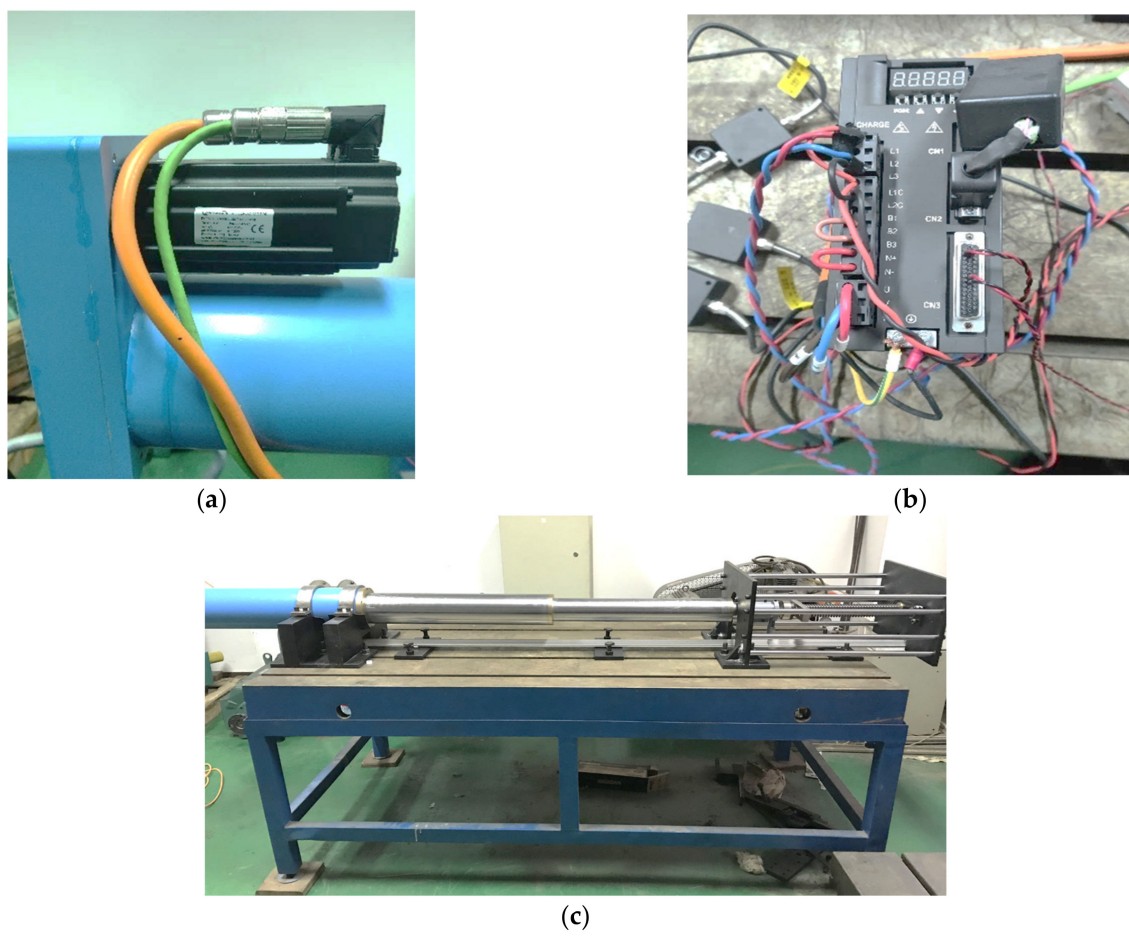

(**a**)

(**b**)

(**c**)

**Figure 16.** (**a**) PMSM; (**b**) Motor driver; (**c**) Three-stage actuating cylinder.

In the Figure 17, the lateral force is applied as follows: the spring at ① applies force to the sleeve of the EMA, and the angle of the sleeve moving forward is offset. When the sleeve moves forward, the friction generated by the contact surface at ② and ③ is the source of lateral force.

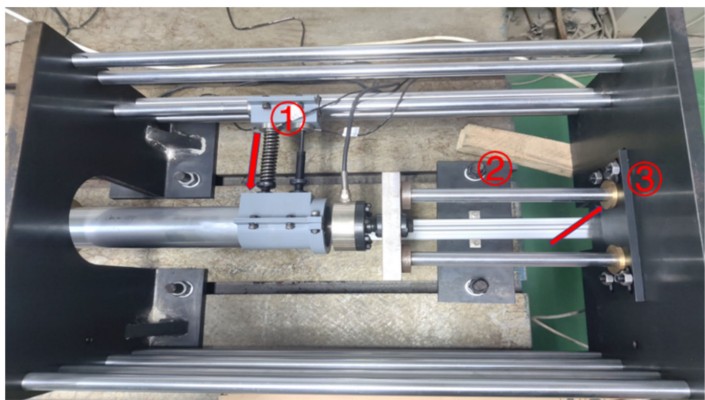

**Figure 17.** Diagram of loading system.

### 4.2. EMA System Experiment

### 4.2.1. Step Command

Given the step command, make the multi-stage electromechanical actuator move from 1450 mm to 1700 mm. During the movement of the electromechanical actuator, a lateral force in the form of Figure 18 is given through mechanical loading.

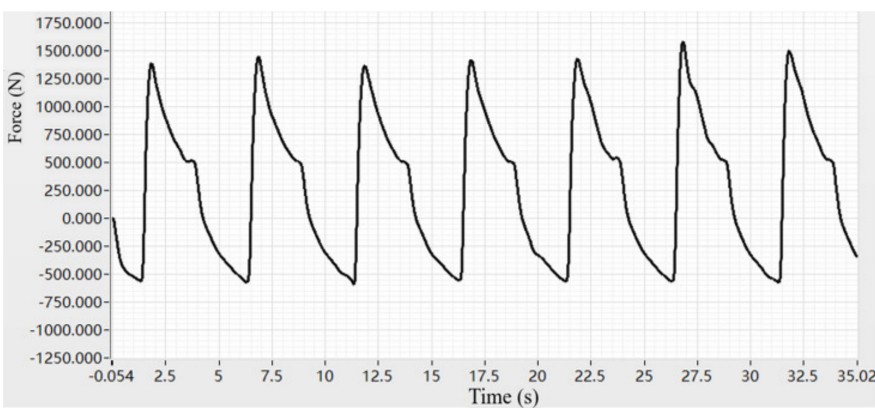

**Figure 18.** Lateral force waveform under step command.

Figure 19a is the EMA displacement curve under the PID control algorithm. It can be obtained that the system reaches a stable state after 6 s. Under the influence of lateral force, the speed and stability of the system following the step command are poor, and the displacement fluctuation is relatively large. Figure 19b is the displacement curve of multi-stage EMA using ADRC algorithm. The time for the system to reach stability is about 4.5 s, and there is no obvious fluctuation in the displacement curve under the action of lateral force, indicating that the motion process of EMA is relatively smooth.

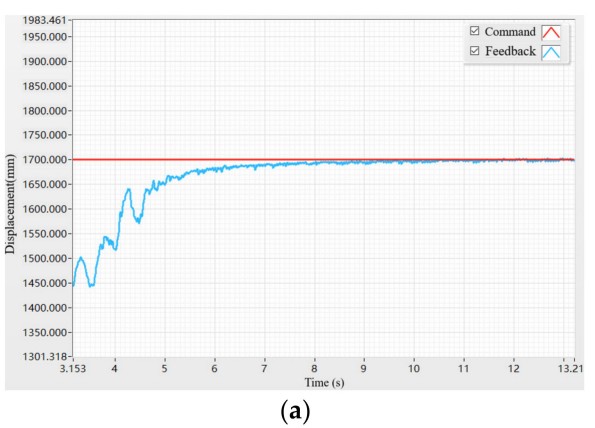
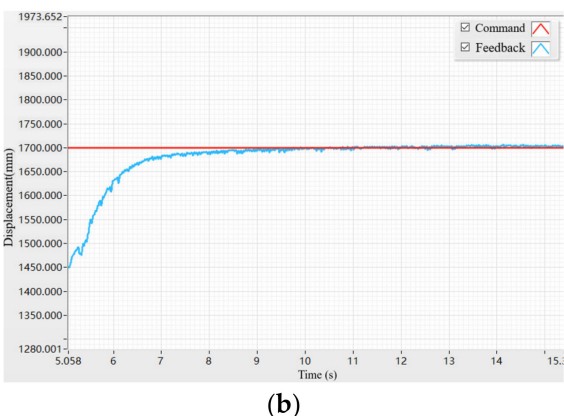

(**a**)　　　　　　　　　　　　　　　　　　(**b**)

**Figure 19.** (**a**) Step response displacement curve of PID; (**b**) Step response displacement curve of ADRC.

Table 2 shows the comparison of time domain indexes between traditional PID algorithm and ADRC algorithm. From the data in the table, it can be concluded that auto-disturbance rejection algorithm can effectively suppress the lateral force interference of multi-stage EMA in linear motion and make the system run smoothly.

**Table 2.** Time domain index under step command.

| Algorithm | Adjustment Time (s) | Time of Displacement Fluctuation (s) |
|:---:|:---:|:---:|
| PID | 6 | 2 |
| ADRC | 4 | 0.2 |

### 4.2.2. Sinusoidal Command

Given the sinusoidal command with frequency $f = 0.1$ Hz, make the multi-stage EMA move back and forth in 1400 mm~1700 mm. At the same time, during the movement of EMA, a lateral force as shown in Figure 20 is given through mechanical loading.

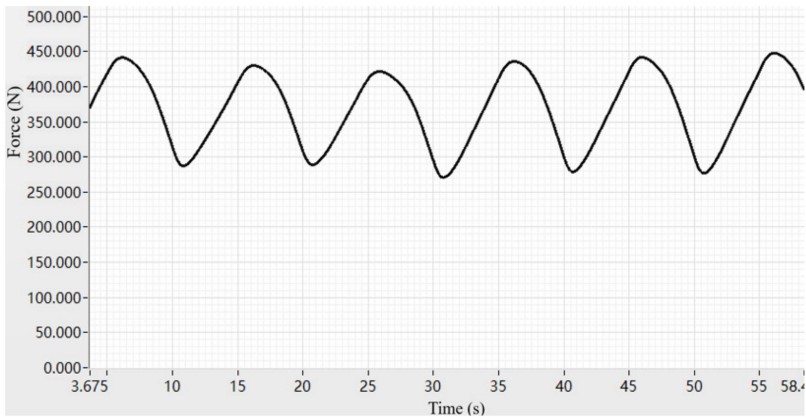

**Figure 20.** Lateral force waveform under sinusoidal command.

Figure 21a is the EMA displacement curve under the PID control algorithm. It can be seen that under the influence of lateral force, the system has poor rapidity and stability to follow the step command, and the displacement jitter is relatively large, which cannot effectively suppress the interference caused by the lateral force. Figure 21b is the EMA displacement curve under the ADRC algorithm. From the diagram, it can be seen that the system has sufficient rapidity and stability to follow the step command, with a lag time of about 0.4 s, and with small displacement fluctuation and strong anti-interference.

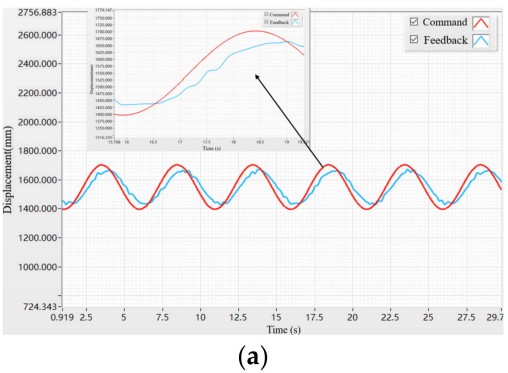 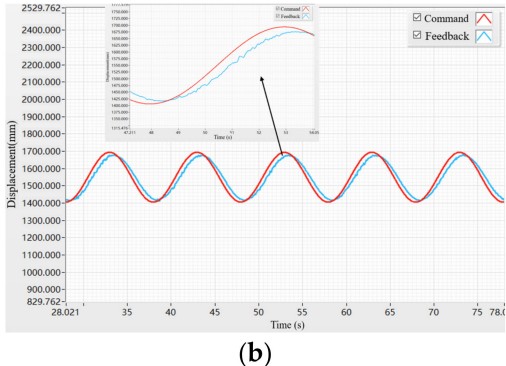

|  (a) | (b) |

**Figure 21.** (**a**) Sine response displacement curve of PID; (**b**) Sine response displacement curve of ADRC.

Table 3 shows the comparison of time domain indexes between traditional PID algorithm and the ADRC algorithm. When EMA isunder the lateral force interference, the multi-stage EMA system under the active disturbance rejection algorithm has better tracking performance. Under the same command, both lag time and amplitude attenuation are better than the traditional PID strategy.

**Table 3.** Time domain index under sinusoidal command.

| Algorithm | Lag Time (s) | Attenuation Amplitude |
|-----------|--------------|------------------------|
| PID | 0.5 | 2.67% |
| ADRC | 0.4 | 1.33% |

### 4.2.3. Random Lateral Force

After the system is completely stable, the lateral force is suddenly added at the uncertain time. Through mechanical loading, the static friction and dynamic friction coupling between the lead screw end and the loading end are used to form the interference torque with uncertain period and size, so as to reasonably simulate the random characteristics of wind load. The lateral force is shown in Figure 22.

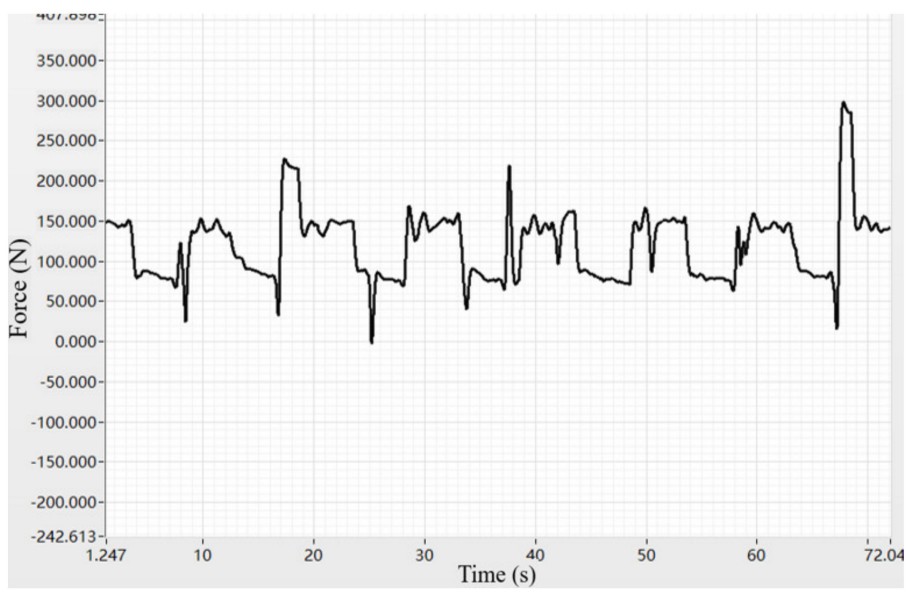

**Figure 22.** Lateral force waveform under random command.

Figure 23a is the displacement curve of the EMA with PID control algorithm after being affected by the lateral force. It can be seen from the figure that the displacement of the EMA returns to stability after 6 s, and the displacement fluctuation amplitude is ± 70 mm. Figure 23b is the lead screw speed curve. From Figure 23c, it can be obtained that the lead screw speed can return to the stable value after about 10 s, and the maximum amplitude during fluctuation is 120 mm/s. As shown in Figure 23b, the displacement of the EMA with ADRC algorithm can be restored to stability after being affected by the lateral force after 1.6 s, and the displacement fluctuation amplitude is ±12 mm. Figure 23d is the lead screw speed curve. From the diagram, it can be obtained that the lead screw speed can be restored to a stable value after about 1.6 s, and the maximum amplitude during fluctuation is ±80 mm/s.

Table 4 shows the effects of the two control algorithms when sudden lateral force is applied after the system is stable. From the table, it can be concluded that the ADRC algorithm can effectively suppress the sudden lateral force and quickly reach a new stable state.

**Table 4.** Time domain index under random lateral force.

| Algorithm | Displacement Recovery Time (s) | Displacement Fluctuation Amplitude (mm) | Speed Recovery Time (s) | Average Velocity Fluctuation Amplitude (m$^2$/s) |
|---|---|---|---|---|
| PID | 6 | ±70 | 10 | ±120 |
| ADRC | 1.6 | ±12 | 1.6 | ±80 |

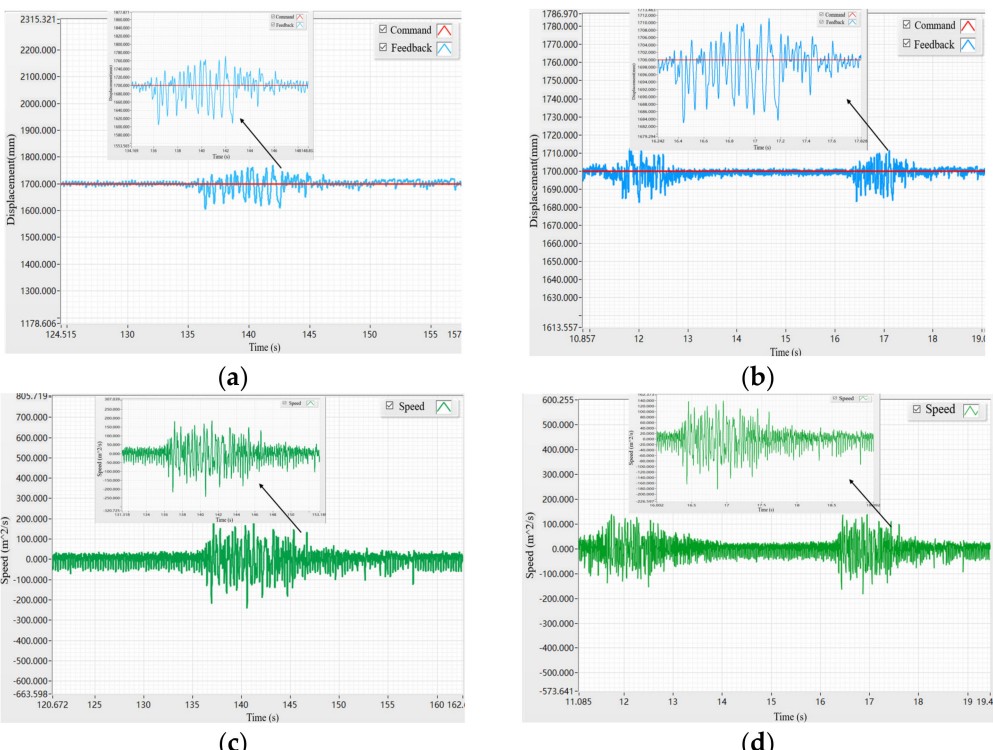

**Figure 23.** (**a**) Displacement curve under stable loading of PID; (**b**) Displacement curve under stable loading of ADRC; (**c**) Speed curve under stable loading of PID; (**d**) Speed curve under stable loading of ADRC.

## 5. Conclusions

Taking the EMA as the research object, this paper designs a stability control simulation model of a multi-stage EMA and a dynamic model of lateral loading force by using the co-simulation of different software, and studies the stability control under the influence of lateral force combined with the improved ADRC technology. The simulation results and experiments show that the designed control algorithm has better command tracking ability, anti-interference ability, and better servo performance than the traditional PID control algorithm. At the same time, compared with the traditional mathematical model, the collaborative simulation model is closer to the actual working condition. It can consider the nonlinear factors in EMA better and provides a solution for the mismatch of control paraments between simulation and engineering. Incidentally, this o-simulation method has a wide range of applications and has reference significance in the modeling and simulation of other systems.

**Author Contributions:** S.W., Y.Z., Y.L. were in charge of the whole trial; S.W. wrote the manuscript; J.Z. and S.M. assisted with sampling and laboratory analyses. All authors have read and agreed to the published version of the manuscript.

**Funding:** This research was funded by Key R & D projects in Shaanxi Province, grant No. 2021ZDLGY10-08, and National Natural Science Foundation of China, grant No. 51875458, 51905428.

**Data Availability Statement:** Data sharing is not applicable.

**Conflicts of Interest:** The authors declare no conflict of interest.

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
