# Peer review of "Research on Stability Control Method of Electro-Mechanical Actuator under the Influence of Lateral Force"

_electronics, doi:10.3390/electronics11081237_

Round 1
Reviewer 1 Report
The paper presents the analysis of the lateral force of the electro-mechanical actuator in several stages. The results of the simulations and the experiment are presented. The paper is rich in information and brings some contributions, especially in terms of the mechanical part of the actuator.
Some comments are given below:
In order for the paper to be published in Electronics, the electrical parameters of the actuator should be analyzed much more.
The conclusions must contain more important information from the paper.
The conclusions should also relate to the electrical characteristics analyzed.
Line 96: Please delete "2".
In Table 1: Correct "6 kW", "50Nm".
In the paper the unit of speed is written in several forms: rpm; rev / min; r / min. Use only one.
In Figure 3b: What is represented on the ordinate and the abscissa?
Clarity can be improved in Figure 3, Figure 4b, Figure 5, Figure 13, Figure 14 and other graphs / figures. Lowercase letters are difficult to read.
Line 100: Subheading 2 is not written.
All the terms must be explained, from relations 1, 2, 4, 5, 6 and the others.
Please correct: "kg.m^2" (in line 234) and "m^2/s" (in table 4).
In figures 13 and 14 what is represented on the ordinate?
I don't think the title of Figure 17 matches the image.
Reviewer 2 Report
This is an interesting paper which presents stability control method of multi-stage Electro-mechanical actuator affected by wind lateral force.
Here are some comments:
- The abstract should more precisely highlight the novelty and aim of this study.
- Simulation platforms which are used in the research and first time mentioned in abstract should be briefly explained/defined.
- Selection of design motor parameters could be better explained.
- Conclusion should be broader and better present novelty and applicability of findings.
- Figure 1. – drawing position definition– please arrange them better
- Lines 66-67 – please consider better formulation of the sentence „In special application field..“
- Line 96 – typing error
- Line 110 -please check sentence “Fourier analysis…”
- Line 120 – Uppercase letter in Figure 4
- Line 128 and 129 – please check
- Please check the Numbering of paragraphs , you have two times 2.3 (line 147 and 180)
Also lines 207, 251, 294, 314, 335
- Please check lines 188 – 192 and the typing style for explain wind load from the Figure 8.
Reviewer 3 Report
The manuscript presents a research on stability control method of electro-mechanical actuator under the influence of lateral force . The following comments are given to further improve the manuscript quality:
1) The authors should be more clear and better stress the novelty of their work.
2) Avoid lumping references, e.g. 15-18 and similar. Instead summarize the main contribution of each referenced paper in a separate sentence and/or cite the most recent and/or relevant one.
3) The authors should more clearly explain why they use the model based on JMAG and why such a model turns out to be more suitable for this particular application.
In overall the contribution of the manuscript is satisfactory but it still needs a minor revision.
Round 2
Reviewer 1 Report
The paper has been improved. Some clarifications and corrections were made in the revised version. I consider that the paper can be published. However, minor corrections still need to be made.
In lines 318, 338 and 359, the titles should be numbered as follows: 4.2.1., 4.2.2., 4.2.3.
Figure 17 has the same title, it needs to be changed. Changes and additions to this figure are written in the "Author Response File", but were not included in the revised paper.
Also, in the titles of figures 18, 20 and 22, the expression "schematic diagram" is not correct.
Author Response
Please see the attachment.
This manuscript is a resubmission of an earlier submission. The following is a list of the peer review reports and author responses from that submission.